# Establishment and Evaluation of Nomogram Model for Predicting the Risk of Arteriovenous Fistula Dysfunction in Patients Undergoing MHD

**DOI:** 10.3390/healthcare13233161

**Published:** 2025-12-03

**Authors:** Dan Jiang, Ling Sun, Minghui Wang, Yahui Han, Youfen Liao, Ling Wang, Xia Fu

**Affiliations:** 1Department of Nephrology, The Eighth Affiliated Hospital, Sun Yat-sen University, Shenzhen 518033, China; jiangd26@mail.sysu.edu.cn (D.J.); sunling3@mail.sysu.edu.cn (L.S.); wangmh88@mail.sysu.edu.cn (M.W.); hanyh35@mail.sysu.edu.cn (Y.H.); liaoyf35@mail.sysu.edu.cn (Y.L.); 2Faculty of Medicine, Macau University of Science and Technology, Macau 999078, China

**Keywords:** arteriovenous fistula, hemodialysis, dysfunction, nomogram, predictive mode

## Abstract

**Background/Objectives:** We aimed to construct a nomogram model for predicting arteriovenous fistula dysfunction risk and to conduct internal validation. **Methods:** The clinical data of 335 patients from the 8th Affiliated Hospital of Sun Yat-Sen University, collected from January 2019 to January 2024, were retrospectively analyzed. Among these patients, 103 were assigned to the arteriovenous fistula (AVF) dysfunction group, while 232 were in the non-dysfunction group. In this study, we first identified risk factors for AVF dysfunction using univariate and logistic regression analyses, and then constructed a prediction model by resampling the data. The model’s performance was evaluated using the C-index, ROC curve, calibration plot, and decision curve analysis, confirming its strong predictive ability and clinical value. **Results:** The results indicated that post-dialysis hypotension, abnormal fibrinogen levels, platelet abnormalities, total cholesterol levels, and diabetes mellitus emerged as independent risk factors for AVF dysfunction in MHD patients; however, total protein levels were a protective factor for AVF dysfunction. The model’s performance was assessed using the receiver operating characteristic (ROC) curve, the Hosmer–Lemeshow test, and the calibration curve. The ROC curve results demonstrated that the area under the curve (AUC) for the training set was 0.852 (0.799–0.904), while that for the validation set was 0.810 (0.715–0.906), indicating good calibration. The decision curve analysis revealed that the predictive nomogram was clinically useful when the threshold for intervention was set between a 15% and 78% probability of dysfunction. **Conclusions:** The nomogram prediction model constructed in this study can be used to predict the risk of autogenous arteriovenous fistula dysfunction in hemodialysis patients.

## 1. Introduction

Hemodialysis is the most widely applied therapy for individuals suffering from end-stage renal disease. Arteriovenous fistula (AVF) is often used as a vascular access option in clinical settings, serving as a critical lifeline for patients undergoing hemodialysis and featuring a high rate of clinical application [1]. Arteriovenous fistula malfunction leads to inadequate blood circulation during dialysis sessions, which can worsen patients’ health conditions and might even increase the danger of further complications, exerting a notable influence on the safety of dialysis therapies. Research suggests that the patency rate of AVF over a five-year period varies between 50.5% and 80.5% [2,3]. As a result, arteriovenous fistula failure has become the main cause for readmission among patients receiving maintenance hemodialysis (MHD), highlighting the significance of promptly detecting and preventing arteriovenous fistula failure [4,5].

At present, there is no dependable and robust predictive model that can be used to identify patients at high risk and prevent the occurrence of arteriovenous fistula dysfunction. Lacking an effective predictive tool brings great challenges to clinical staff in advance intervention and risk management of arteriovenous-fistula-related problems. Most models used for predicting clinical events make extensive use of nomogram modeling, which is recognized as a trustworthy statistical tool in the field of medical research [6]. Specifically, a nomogram is a visual line chart with intuitive expression that integrates a variety of clinical indicators with multivariate regression analysis technology. Through this combination, it can accurately predict the probability of a specific clinical outcome [7,8]. The visual characteristic of a nomogram makes it easy for clinical workers lacking in-depth statistical knowledge to understand and apply, which is an important reason for its wide application in clinical event prediction.

Regarding the topic of AVF, Zhao B. et al. [9] identify factors that may predict arteriovenous fistula maturation in patients undergoing maintenance hemodialysis; Guo S. et al. [10] analyze the factors associated with AVF failure in MHD patients and establish a nomogram prediction model; and Liu S. et al. [11] identify risk factors contributing to premature AVF maturation in elderly diabetic patients and develop a clinical prediction model. Given the significant differences in the characteristics of MHD patients, the artificial blood vessel utilization rates both domestically and internationally, and the occurrences of internal fistula loss in MHD patients in China, fewer nomogram models have been developed to predict arteriovenous fistula dysfunction. Therefore, we constructed and validated a nomogram model based on relevant variables.

To develop a nomogram for the quantitative evaluation of AVF dysfunction risk in MHD patients, in this research, we conducted a retrospective analysis of MHD patients using accumulated data from the hemodialysis information system. The aim was to identify high-risk factors, thereby providing a reference for the clinical prediction and further guiding the improvement of AVF dysfunction in MHD patients.

## 2. Materials and Methods

### 2.1. Study Population and Design

We retrospectively analyzed 335 patients undergoing maintenance hemodialysis (MHD) who were admitted to the 8th Affiliated Hospital of Sun Yat-sen University between September 2014 and October 2024. The patients’ ages ranged from 20 to 90 years, with a mean age of (58.1 ± 14.4) years; 200 were male, and 135 were female; and the primary pathologies included primary glomerulonephritis in 76 patients, diabetic nephropathy in 102, hypertensive nephropathy in 98, and other causes in 59. Overall, this study included 335 patients, with 235 in the modeling group and 100 in the validation group (Figure 1). This study was conducted in accordance with the Declaration of Helsinki, approved by the Biomedical Ethics Committee of the Eighth Affiliated Hospital of Sun Yat-sen University (IRB No. 2025-06-03), and informed consent was obtained from all patients.

### 2.2. Methods

The entire cohort was randomly split into training and validation sets at a 7:3 ratio using R (https://www.r-project.org), with a sample size of 235 for the training group and 100 for the validation group. Clinically relevant variables identified in the comparative analysis were subjected to univariate and multivariate logistic regression to identify independent predictors of AVF dysfunction. Stepwise regression based on the minimum Akaike Information Criterion (AIC) was performed in R to select the final predictors for the AVF dysfunction model. Using the ‘rms’ package, a nomogram was developed based on these final predictors, the predictive performance of which was assessed using the area under the receiver operating characteristic curve (AUC), calibration plots, and decision curve analysis (DCA), with *p* < 0.05 considered statistically significant. The model demonstrated poor goodness of fit.

### 2.3. Criteria for Selection

Inclusion criteria: (1) end-stage renal disease; (2) AVF established for more than 6 months; and (3) hemodialysis via AVF. Exclusion criteria: (1) cardiac factors leading to a decrease in the blood flow rate controlled by the dialysis pump and (2) insufficient data.

### 2.4. AVF Functional Assessment: Methodology and Criteria

AVF dysfunction was defined as meeting one or more of the following criteria: (1) blood flow rate < 200 mL/min during dialysis; (2) ≥50% reduction in vascular diameter; and (3) weakened thrill or thrombus formation. Patients fulfilling any of these criteria were classified into the dysfunction group [12].

### 2.5. Observational Indicators

#### 2.5.1. Data Collection

We used the hospital medical record system information platform to collect general data including gender, age, body mass index (BMI), dialysis duration, and presence of other underlying diseases (diabetes, hypertension, coronary heart disease).

#### 2.5.2. Laboratory Indicators

We used the hospital medical record system information platform to obtain key laboratory parameters, including common hematological indices such as the levels of hemoglobin (Hb), triglycerides (TGs), cholesterol (TC), low-density lipoprotein cholesterol (LDL-C), and high-density lipoprotein cholesterol (HDL-C), high-sensitivity C-reactive protein (hs-CRP), fibrinogen (Fib), etc. Blood samples from the AVF dysfunction group were collected during routine examinations, while those from the non-dysfunction group were obtained during hospital admission. Peripheral venous blood samples (5 mL) were obtained from all patients prior to hemodialysis sessions.

### 2.6. Statistical Analysis

Data were processed using SPSS 26.0 software, with count data represented as [n (%)] and subjected to row Chi-square tests and measurement data presented as (x ± s) and subjected to row *t*-tests. All laboratory data were extracted through the hospital’s integrated platform. Following data collection, two researchers independently compiled and cross-verified the records, eliminating samples with apparent logical inconsistencies or questionable authenticity to ensure data accuracy and reliability. Variables with missing values exceeding 20% were excluded from the final dataset, while those with ≤20% missingness were handled using mean imputation.

To develop the nomogram, we compared patient characteristics between the AVF dysfunction and normal function groups and performed multivariate logistic regression to identify independent predictors of AVF non-dysfunction. Finally, odds ratios (ORs) with 95% confidence intervals (CIs) were calculated for each risk factor in the logistic regression model.

The nomogram’s performance was validated in the training and validation cohorts. Discriminative capacity was quantified by the area under the receiver operating characteristic curve (AUC), with a value of 1.0 representing ideal discrimination. Calibration was examined through the Hosmer–Lemeshow test (*p* > 0.05), showing satisfactory agreement between predictions and observations. Further assessment via calibration curves visualized the concordance between predicted and actual probabilities across both cohorts.

## 3. Results

### 3.1. The Baseline Data and Clinical Indicators of Patients in the Modeling Group

The patients’ clinical information is summarized in Table 1. The modeling group consisted of 335 patients, with 232 in the non-dysfunction and 103 in the AVF dysfunction group. A comparative analysis of both groups in terms of general clinical data, hematological indicators, and surgical indicators revealed no statistically significant differences in the primary etiology of the disease, height, sex, HDL, LDL, calcium, phosphorus, albumin, glucose, serum parathyroid hormone, serum potassium, platelet count, hematocrit, serum BUN, serum creatinine, aspartate transaminase, alanine transaminase, total iron-binding capacity, triglycerides, and total cholesterol (all *p* > 0.05). However, there were statistically significant differences between the groups (all *p* < 0.05) in CO_2_CP, uric acid, total protein (TP), diabetes, hs-CRP, platelet count (PLT), parathyroid hormone, LDL, HDL, fibrinogen, total cholesterol, serum calcium–phosphorus product, and hypotension after dialysis, as shown in Table 1.

### 3.2. Collinearity Assessment

Based on the univariate analysis results (*p* < 0.02), variables meeting the significance threshold were selected for collinearity assessment. Using linear regression models, multicollinearity was evaluated through variance inflation factor (VIF) and Tolerance values, with VIF < 10 and Tolerance > 0.1 defined as acceptable thresholds. Finally, seven key predictors (TC, total protein, diabetes, Hs-CRP, PLT, fibrinogen, serum (Ca-P) product, and hypotension) were identified by integrating statistical collinearity diagnostics with clinical relevance in Table 2.

### 3.3. Analysis of Risk Factors for Arteriovenous Fistula Dysfunction in the Modeling Group

Seven key predictors (abnormal PLT, TC, diabetes, abnormal Fib, abnormal (Ca-P), hypotension after dialysis, TP) were identified as independent factors for AVF dysfunction among MHD patients by integrating statistical results from univariate and collinearity analyses with clinical relevance, a binary logistic regression analysis was conducted. The assignment of the independent variable values is presented in Table 3. The results indicated that abnormal PLT (OR = 4.18, 95% CI: 1.95–8.99), TC (OR = 1.71, 95% CI: 1.31–2.24), diabetes (OR = 3.20, 95% CI: 1.77–5.77), abnormal Fib (OR = 2.65, 95% CI: 1.46–4.80), abnormal (Ca-P) (OR = 1.81, 95% CI: 1.00–3.27), and hypotension after dialysis (OR = 4.14, 95% CI: 2.31–7.42). TP (OR = 0.93, 95% CI: 0.89–0.97) was a protective factor for AVF dysfunction among MHD patients according to multivariate logistic regression analysis (*p* < 0.05, Table 3 and Table 4, and Figure 2).

### 3.4. Establishing a Predictive Model for AVF Dysfunction

Six predictors are integrated into the model: PLT, diabetes, Fib, Ca-P product, hypotension after dialysis, and TC. Points are assigned to each variable, and the sum yields a total score indicative of the failure risk. For example, a score of 64 points, calculated from the respective contributions of each factor, is directly mapped to a patient’s probability of experiencing AVF dysfunction, providing a readily applicable clinical tool, as shown in Figure 3.

### 3.5. Internal Validation of the AVF Dysfunction Risk Prediction Model

The dataset utilized for modeling was randomly split into a training and validation subset at a ratio of 7:3. To be precise, the training subset comprised 235 samples, while the validation subset included 100 samples.

#### 3.5.1. Discriminative Ability Assessment

The analysis of the receiver operating characteristic (ROC) curve revealed that the area under the curve (AUC) for the training subset was 0.852 (0.799–0.904), whereas that for the validation subset was 0.810 (0.715–0.906). These results indicated that the model possessed a strong discriminative ability, effectively distinguishing between samples with different outcomes. Furthermore, we compared the model’s predictive capabilities for training and validation (Table 5; Figure 4).

#### 3.5.2. Calibration Performance Assessment

The calibration curve analysis revealed no notable discrepancy between the risk probabilities forecasted by the nomogram and the actual target event occurrence rates. This outcome indicated that the model exhibited excellent calibration performance, meaning the predicted risk values could accurately reflect the real likelihood of event occurrence (Figure 5A,B).

#### 3.5.3. Clinical Applicability Assessment

Decision curve analysis (DCA) was utilized to evaluate the prediction model’s clinical applicability. When the threshold risk probability ranged from 15% to 78%, the net benefit obtained by patients through the model exceeded that of the two extreme curves in the graph, typically representing the strategies of “providing intervention to all patients” and “providing no intervention to any patient.” This observation confirmed that, within this risk threshold range, the model could deliver high clinical treatment effectiveness when guiding clinical decisions (Figure 6A,B).

## 4. Discussion

AVF serves as a critical lifeline for patients with end-stage renal disease, and maintaining its optimal function is essential for those undergoing hemodialysis [13]. As a prognostic prediction tool, the nomogram [14] has gained widespread application across various fields. This is primarily due to its user-friendly visual display, simple numerical calculation process, and ability to enhance clinical decision-making efficiency [15,16,17,18]. In this study, we developed a nomogram prediction model to identify the risk of AVF dysfunction at an early stage. The model demonstrated excellent discriminative ability, with AUC values of 0.852 in the training cohort and 0.810 in the validation cohort. The Hosmer-Lemeshow test and calibration curves confirmed a good model fit. Decision curve analysis indicated that the model provided clinical benefit when the dysfunction risk threshold fell within the 15–78% range. If interventions are implemented early at a risk probability of 14%, AVF dysfunction risk can be effectively reduced, thereby supporting clinical decision-making and improving prognosis.

In this study, we developed and validated a nomogram prediction model for AVF dysfunction in patients with chronic kidney disease undergoing MHD, identifying key factors influencing AVF dysfunction through risk factor screening and the establishment of a nomogram prediction model. These factors encompass post-dialysis hypotension, abnormal fibrinogen count, abnormal PLT count, TC, and diabetes. Previous research found that hemodialysis patients with diabetes experience vascular intimal hyperplasia and thickening, which are associated with a higher risk of AVF dysfunction [19,20], a finding corroborated in this study. Additionally, abnormal platelet counts and fibrinogen levels may induce a chronic low-grade inflammatory state in blood vessels or alter endothelial structure, leading to stenosis, occlusion, or thrombosis [21,22]. Hypercholesterolemia contributes to AVF dysfunction in MHD patients by promoting atherosclerosis, thrombosis, and vascular intimal hyperplasia, making it a significant and modifiable risk factor, in alignment with the results of this study [23,24,25]. Furthermore, post-dialysis hypotension is also associated with AVF dysfunction, possibly due to reduced blood flow and velocity through the fistula during hypotensive episodes [26,27]. Unlike previous studies, it is worth mentioning that in this study, we found that TP was a protective factor for AVF dysfunction in hemodialysis patients, likely due to higher total protein levels in these patients that may reflect better nutritional status and vascular elasticity [28,29,30].

Current risk prediction models for hemodialysis patients primarily focus on forecasting early [31] and late-stage [32] AVF dysfunction. In contrast, the mature autogenous arteriovenous fistula failure prediction model developed in this study demonstrates superior applicability; compared to existing models, our model encompasses a broader patient population with wider age distribution, utilizes readily obtainable risk factors, and offers enhanced convenience for clinical implementation and dissemination. Furthermore, in this research we have digitized the risk prediction model by integrating a fistula failure prediction module into the hemodialysis information system, enabling nursing staff to assess patients’ fistula failure risk directly through the system and promptly implement clinical interventions.

However, this study has several limitations that should be acknowledged. First, the model was developed based on a single-center cohort, which may limit its generalizability. Second, due to data constraints, genetic predisposition and self-management capacity were not considered in the analysis, which may affect the comprehensiveness of the model. Third, it should be noted that the cohort included 103 patients with arteriovenous fistula dysfunction and 232 with normal function, resulting in a ratio of approximately 1:2. This represents only mild class imbalance; therefore, no data balancing methods were applied. Nevertheless, severe class imbalance could lead to model bias toward the majority class and reduce predictive accuracy for minority class instances. In light of these limitations, researchers in future studies should incorporate more comprehensive influencing factors and adopt multi-center designs to further refine and validate the model, not only enhancing its accuracy and applicability but also providing more targeted guidance for clinical practice.

## 5. Conclusions

In summary, in this study, we have preliminarily established a nomogram prediction model that can provide clinicians with theoretical references when assessing AVF dysfunction risk. Early measures should be taken in terms of monitoring risk factors and medical intervention to reduce AVF dysfunction risk. The next step in this research is to promote multi-center collaboration, continue to evaluate and improve the prediction model, and enhance its accuracy.

## Figures and Tables

**Figure 1 healthcare-13-03161-f001:**
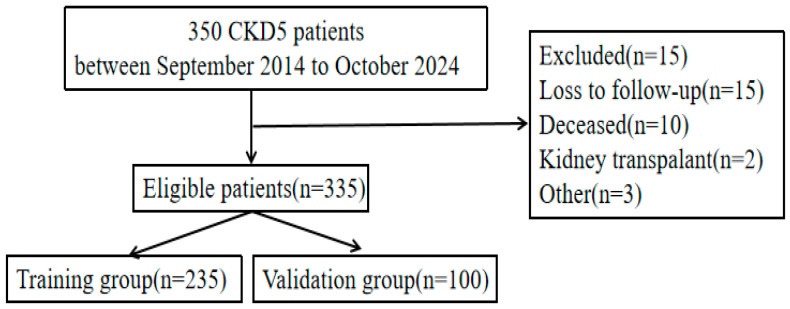
Flowchart depicting the inclusion of CKD5 patients in this study.

**Figure 2 healthcare-13-03161-f002:**
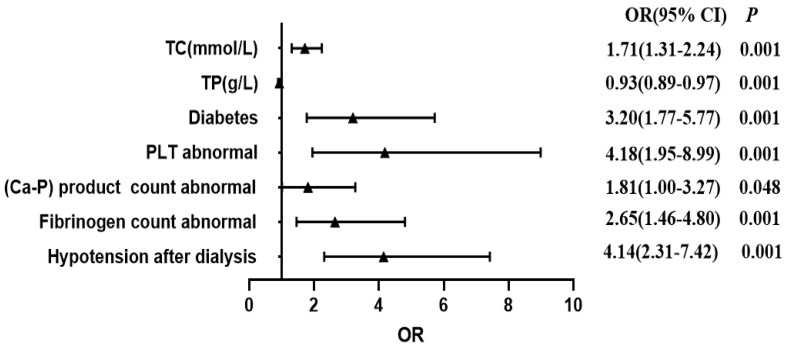
Multi-factor logistic regression analysis of AVF dysfunction risk factors. Forest plot of the multivariate logistic analysis. Independent variables with *p* < 0.05 PLT abnormal, platelet abnormal; TC, total cholesterol; (Ca-P) product count abnormal, calcium–phosphorus product abnormal; TP, total protein; Diabetes; Fibrinogen abnormal levels were screened with the backward method to construct the model.

**Figure 3 healthcare-13-03161-f003:**
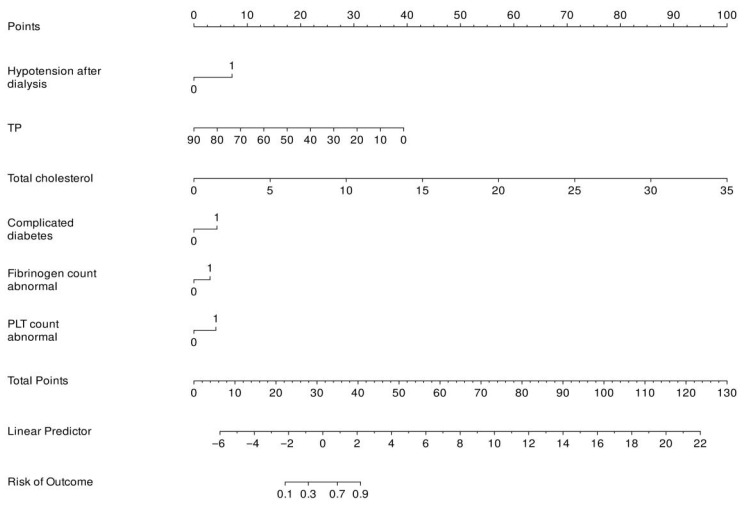
Nomogram model for predicting AVF dysfunction risk.

**Figure 4 healthcare-13-03161-f004:**
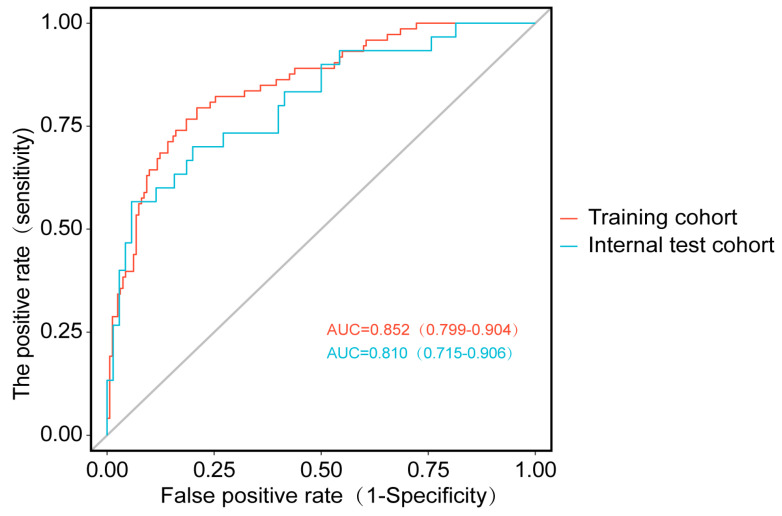
AUC of the training and validation sets. The Area Under the Curve (AUC) quantifies model discrimination by summarizing the overall performance of the ROC curve. A perfect classifier achieves an AUC of 1, corresponding to the top-left corner of the plot, while an AUC of 0.5 indicates predictions no better than random chance. Furthermore, the ROC curve itself plots the True Positive Rate against the False Positive Rate across classification thresholds and enables identification of optimal operating points through explicit visualization of the sensitivity and specificity tradeoff.

**Figure 5 healthcare-13-03161-f005:**
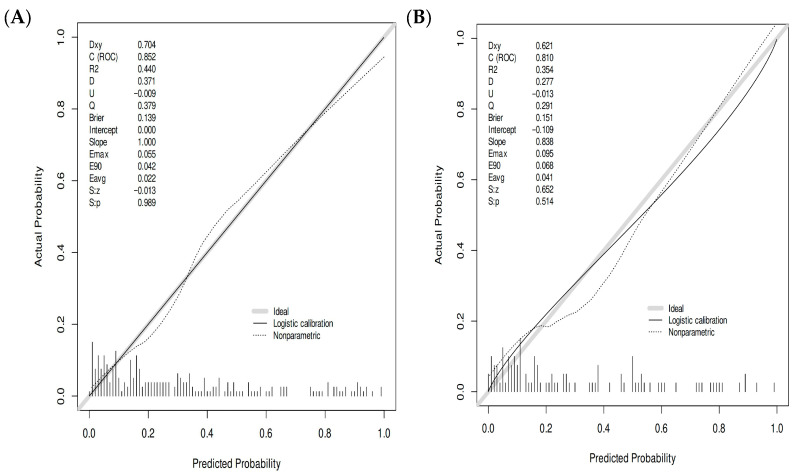
Calibration curve of AVF dysfunction prediction model of the model. Calibration plots of the predictive model. (**A**) Calibration curve of the model in the training set; (**B**) Calibration curve of the model in the internal validation set; The calibration curve displays the predicted probability of AVF dysfunction on the *x*-axis and the actual diagnosed risk of AVF dysfunction on the *y*-axis. The diagonal dashed line in the graph represents the ideal prediction model, while the solid black line represents the risk nomogram prediction model developed in this study. When the solid black line closely approximates the diagonal dashed line, it indicates better predictive performance of the model. As shown in the graph, the developed risk nomogram prediction model for AVF dysfunction demonstrates good predictive accuracy.

**Figure 6 healthcare-13-03161-f006:**
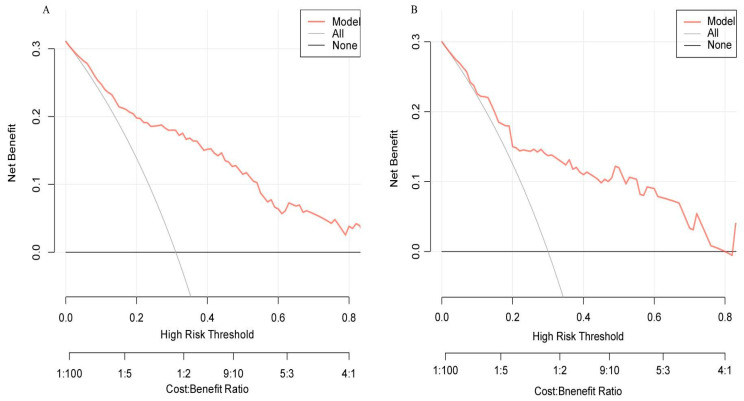
Decision curve analysis for AVF dysfunction prediction model. (**A**) Decision curve of the test set. (**B**) Decision curve of the validation set. The decision tree model demonstrates superior net benefit across a threshold probability range of 0.15 to 0.78, as its DCA curve (red) remains above the “ALL” (grey) and “NONE” (black) reference strategies. The “ALL” model classifies all cases as positive, while the “NONE” model classifies all as negative.

**Table 1 healthcare-13-03161-t001:** Baseline information for the AVF dysfunction and normal function groups.

Category		Dysfunction Group (n = 103)	Non-Dysfunction Group (n = 232)	t/Z/χ^2^	*p*
Age (years)		57.75 ± 13.66	58.20 ± 14.78	−0.266	0.790
Height (cm)		164.08 ± 0.90	165.62 ± 7.93	−1.649	0.100
Sex	Male	64 (62%)	136 (59%)	0.366	0.545
	Female	39 (38%)	96 (41%)		
CO_2_CP (mmol/L)		23.09 ± 4.18	21.19 ± 3.10	4.615	0.001
Uric acid (mmol/L)		455.92 (388.89, 529.12)	396.91 (315.94, 496.29)	−6.170	0.001
Total Protein (g/L)		64.84 ± 10.59	67.54 ± 5.39	−3.093	0.002
HDL (mmol/L)		1.05 ± 0.24	0.99 ± 0.29	1.937	0.054
LDL (mmol/L)		2.17 ± 0.63	2.10 ± 0.71	0.779	0.437
Hs-CRP (mg/L)		10.20 ± 18.59	6.59 ± 13.25	2.018	0.044
PTH (pg/mL)		266.66 ± 90.90	220.93 ± 160.06	2.271	0.024
Calcium (mmol/L)		2.26 ± 0.21	2.24 ± 0.19	0.550	0.583
Phosphorus (mmol/L)		1.91 ± 0.58	1.97 ± 0.58	−0.953	0.341
Albumin (g/L)		37.67 ± 5.05	37.079 ± 4.10	1.130	0.259
Glucose (mmol/L)		7.53 ± 4.66	7.73 ± 3.89	−0.394	0.694
Potassium (mmol/L)		4.74 ± 0.66	4.71 ± 0.60	0.868	0.386
Hematocrit (%)		0.69 ± 3.365	0.35 ± 0.05	1.570	0.117
BUN (mmol/L)		22.48 ± 8.41	23.31 ± 7.09	−0.930	0.353
Cretine (μmol/L)		976.04 ± 331.89	995.22 ± 286.53	−0.538	0.591
Aspartate transaminase (g/L)		15.69 ± 7.56	16.52 ± 9.01	−0.810	0.419
Alanine transaminase (μ/L)		12.89 ± 6.34	14.51 ± 10.90	−1.403	0.161
Total Iron Binding Capacity (mmol/L)		10.89 ± 4.31	10.95 ± 3.91	−0.133	0.894
Triglyceride (g/L)		2.04 ± 1.613	1.83 ± 1.525	1.156	0.248
Total cholesterol (mmol/L)		3.20 (2.34, 4.01)	3.82 (3.32, 4.30)	−4.682	0.001
LLDL (mmol/L)		2.42 ± 1.68	2.05 ± 0.72	2.847	0.005
DHDL (mmol/L)		1.07 ± 0.35	0.99 ± 0.29	2.297	0.022
Fibrinogen count (g/L)	Normal	31 (30%)	126 (54%)	16.794	0.001
	Abnomal	72 (70%)	106 (46%)		
Complicated diabetes	Yes	55 (54%)	54 (23.3%)	29.485	0.001
	No	48 (46%)	178 (77.0%)		
Platelet count (×10^9^/L)	Normal	31 (30%)	126 (54%)	16.794	0.001
	Abnromal	72 (70%)	106 (45.6%)		
(Ca-P) product count	Normal	31 (30%)	126 (54%)	16.794	0.001
	Abnormal	72 (70%)	106 (46%)		
Hypotension after dialysis	Yes	70 (68%)	77 (33.2%)	35.022	0.001
	No	33 (32%)	155 (66.8%)		

LDL, low-density lipoprotein; BUN, blood urea nitrogen; HDL, high-density lipoprotein; PTH, parathyroid hormone; CO_2_CP, carbon dioxide combining power; Ca-P, serum calcium–phosphorus product.

**Table 2 healthcare-13-03161-t002:** Multicollinearity diagnostic of logistic regression.

Item	Tolerance	VIF
Hypotension after dialysis	0.947	1.056
Fibrinogen abnormal	0.968	1.033
(Ca-P) product count abnormal	0.980	1.021
PLT abnormal	0.989	1.012
Complicated diabetes	0.933	1.072
TP (g/L)	0.995	1.006
TC (mmol/L)	0.972	1.029

PLT, platelet; TC, total cholesterol; Ca-P, serum calcium–phosphorus product; TP, total protein.

**Table 3 healthcare-13-03161-t003:** The assignment of logistic regression.

Item	Assignment
Hypotension after dialysis	1: Yes; 0: No
Fibrinogen abnormal	1: Yes; 0: No
(Ca-P) product count abnormal	1: Yes; 0: No
PLT abnormal	Continuity variables
Complicated diabetes	1: Yes; 0: No
TP (g/L)	Continuity variables
TC (mmol/L)	Continuity variables

**Table 4 healthcare-13-03161-t004:** Multi-factor logistic regression analysis of risk factors of AVF dysfunction.

Item	β	SE	Wald	*p*	OR	95% CI
Hypotension after dialysis	1.421	0.298	22.765	0.001	4.141	2.310–7.422
Fibrinogen (g/L)	0.937	0.304	10.217	0.001	2.645	1.457–4.802
(Ca-P) product count	0.594	0.301	3.899	0.048	1.812	1.004–3.267
PLT (×10^9^/L)	1.431	0.390	13.433	0.001	4.182	1.946–8.988
Complicated diabetes	1.161	0.302	14.810	0.001	3.194	1.768–5.772
TP (g/L)	−0.076	0.022	11.597	0.001	0.927	0.887–0.968
TC (mmol/L)	0.538	0.136	15.570	0.001	1.712	1.311–2.237

**Table 5 healthcare-13-03161-t005:** Prediction probability verification of two groups of early warning models.

Model	Group	AUC	SE	SP	Cut-off	PPV	NPV
LR	training	0.852 (0.799–0.904)	0.866	0.722	0.355	0.646	0.870
	validation	0.810 (0.715–0.906)	0.867	0.768	0.392	0.619	0.930

LR, logistic regression; AUC, area under curve; SE, sensitivity; SP, specificity; PPV, Positive Predictive Value; NPV, Negative Predictive Value; Cut-off, cut-off value.

## Data Availability

The data that support the findings of this study are available from the corresponding author upon reasonable request. The data are not publicly available due to ethical restrictions, as they contain information that could compromise the privacy of research participants.

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
