# Peer review of "Establishment and Evaluation of Nomogram Model for Predicting the Risk of Arteriovenous Fistula Dysfunction in Patients Undergoing MHD"

_healthcare, 2025, doi:10.3390/healthcare13233161_

Round 1

Reviewer 1 Report

Comments and Suggestions for Authors

Research Article Review

Title: Establishment and evaluation of nomogram model for predicting the risk of arteriovenous fistula dysfunction in patients undergoing MHD

The article is about the development and validation of a prognostic model (a type of tool used to predict health outcomes) that can predict the risk of arteriovenous fistula dysfunction in patients undergoing haemodialysis. The topic is important for clinical practice and has real-life applications, as fistula failure is one of the main reasons for dialysis patients having to go back into hospital. The authors looked back at clinical and laboratory data to build a model using something called 'logistic regression analysis'. The results show that the model is good at distinguishing between different types of patients (AUC > 0.8) and that it has potential clinical use.

The article is costructed properly, same changes can help to improve overall quality of presented research. There is necessary to revise some data, especialy in results section.

  1. There is some inaccuracy between tables and text. In Table 1, the 'non-dysfunction' group consists of 232 patients, but in the text it is described as 103 patients.
  2. In Table 1, the sum of females and males in non-dysfunction group is 233.
  3. Table 2 is not presented below text (It is figure 2)
  4. The Discussion section sometimes repeats the Results instead of interpreting them.
  5. The novelty of the work in relation to other studies should be emphasised more strongly.
  6. Text needs linguistic correction (grammar, coherence, sentences too long).
  7. It would be beneficial to include a concise section offering practical advice for clinicians on utilising the nomogram.

I greatly appreciate the authors' contributions. I recommend publication of the article after minor revisions

Reviewer 2 Report

Comments and Suggestions for Authors

The manuscript titled Establishment and evaluation of nomogram model for predict- ing the risk of arteriovenous fistula dysfunction in patients undergoing MHD by Dan Jiang et al presented nomogram prediction 
model to predict the risk of autogenous arteriovenous fistula dysfunction in hemodialysis patient. 

The manuscript is clinically relevant and aim to address an important unmet need. But the manuscript need improvement. 

The manuscript focuses on nomogram model but mehtod section not adequately described. 

Autors used limited predictor factors. Do authors have access to ultrsound, genetic? 

Did authors adjusted for age variable? Did authors analyze the prediction model male female separately?

Some variables show significance in univariate analysis but multivariate not described. 

Units and refence should be consistencely reported. 

All the figure legend should include detail information. 

Some figure needed improvement of the quality. 

Should discuss detail how the model can be useful. 

Repeatitive sentense need to be check. 

Reviewer 3 Report

Comments and Suggestions for Authors

This study addresses a relevant clinical need- predicting AVF dysfunction in MHD patients- and proposes a nomogram-based model with strong potential for early intervention and individualized care. The topic is timely, and the use of multiple validation methods (ROC, calibration curve, DCA) strengthens the manuscript.

However, several areas of the Methods section need clarification. The process for dividing patients into modeling and validation groups should be explicitly described, along with clearer definitions of key clinical variables and risk factors. Using dialysis flow rate as both a group criterion and outcome may introduce bias. This should be rethought or justified. Ethical approval details are also missing and should be added.

Statistical analysis is generally appropriate, but the selection process for variables and handling of missing data need more transparency. Lab and clinical indicators would benefit from standardized units and clearer timing of collection.

The results are logically presented, and the nomogram performs well, with AUC values showing strong predictive ability. The discussion connects well with existing literature and highlights practical implications, though it could be more concise.

Overall, this is a valuable contribution with strong potential impact, but a major revision is needed to improve methodological clarity and precision.

Recommendation: Major Revision

Reviewer 4 Report

Comments and Suggestions for Authors

Peer Review Report

Title: Establishment and evaluation of nomogram model for predicting the risk of arteriovenous fistula dysfunction in patients undergoing MHD

The manuscript presents a well-structured and comprehensive study aimed at developing and validating a nomogram model to predict the risk of arteriovenous fistula (AVF) dysfunction in maintenance hemodialysis (MHD) patients. The topic is clinically relevant, and the paper demonstrates careful data collection and a systematic modeling approach using logistic regression. The model’s calibration and discrimination statistics are clearly reported, and the study provides meaningful insights for clinical practice.

However, several important points should be addressed to improve the rigor and interpretability of the paper. 

  1. Multicollinearity Assessment
    The study includes a relatively large number of independent variables. This raises the possibility of multicollinearity, which could distort regression coefficients and model interpretation.
    It is recommended that the authors perform a multicollinearity diagnostic (e.g., Variance Inflation Factor, VIF). If significant multicollinearity exists, consider applying regularized regression methods, such as Elastic Net logistic regression, to stabilize coefficient estimates.

  2. Model Selection and Statistical Criteria
    The methodology section would benefit from a more explicit explanation of how variables were selected for the final model.
    Please specify the criteria for inclusion/exclusion (e.g., stepwise selection, AIC/BIC, clinical relevance) and the model selection strategy used to determine the optimal set of predictors. Including a brief mathematical formulation or a schematic of the model structure would improve transparency.

  3. Class Imbalance Problem
    The dataset is unbalanced—103 in the non-dysfunction group versus 232 in the dysfunction group. This imbalance can bias model estimation and predictive accuracy.
    The authors should clarify whether any technique was used to address this issue, such as resampling (SMOTE or undersampling), class-weighted regression, or bootstrapped balancing. Discussion on how imbalance may influence the AUC or calibration performance would strengthen the manuscript.

  4. Interpretation of Predictors
    Figure 2 shows that among multiple predictors, total protein (TP) exhibits a negative association with AVF dysfunction, indicating a protective effect.
    The authors should elaborate on the biological or clinical rationale for this finding. For instance, higher protein levels might reflect better nutritional status and vascular resilience, as suggested in related studies. A short explanatory paragraph in the Discussion would provide valuable context.

  5. Validation Metrics and Predictive Performance
    The authors used a sample of 100 patients for model validation and reported AUC values. However, the predictive ability could be described in greater detail.
    Consider reporting additional predictive metrics—such as sensitivity, specificity, positive predictive value (PPV), negative predictive value (NPV)—and complementary error measures such as RMSE or MAPE, to give a fuller picture of model accuracy.

Round 2

Reviewer 2 Report

Comments and Suggestions for Authors

Thank you for improving most of the parts of the manuscript. 

Please improve the figure legends, as I commented before that figure legend should be more detail, not only 1 sentence. Auhtors refering to the supplementary, figure legend detail should be in the main manuscript file with the figure. 

Author Response

Comments 1: Please improve the figure legends, as I commented before that figure legend should be more detail, not only 1 sentence. 

Response 1:Thank you for pointing this out. We agree with this comment. Therefore, we commented the figure legend with the modifications highlighted in yellow . 

1. Mention exactly where in the revised manuscript this change can be found – page 7,1 paragraph, and 198(Table4)-201-204(Figure2).]

Item

β

SE

Wald

P

OR

95% CI

Hypotension after dialysis

1.421

0.298

22.765

0.001

4.141

2.310-7.422

Fibrinogen (g/L)

0.937

0.304

10.217

0.001

2.645

1.457-4.802

(Ca-P) product count

0.594

0.301

3.899

0.048

1.812

1.004-3.267

PLT(x109/L)

1.431

0.390

13.433

0.001

4.182

1.946-8.988

Complicated diabetes

1.161

0.302

14.810

0.001

3.194

1.768-5.772

TP(g/L)

-0.076

0.022

11.597

0.001

0.927

0.887-0.968

TC(mmol/L)

0.538

0.136

15.570

0.001

1.712

1.311-2.237

Table 3. The assignment of logistic regression

Sig: PLT, platelet; TC, total cholesterol; Ca-P, serum calcium–phosphorus product; TP, total protein.

Figure 2. Multi-factor logistic regression analysis of AVF dysfunction risk factors.

Sig: Forest plot of the multivaniate logistic analysis.lindependent variables with p<0.05 (PLT abnormal, platelet abnormal; TC, total cholesterol; (Ca-P) product count abnormal, calcium–phosphorus product abnormal; TP, total protein; Diabetes; Fibrinogen abnormal ) were screened with the backward method to construct the model.

2. Mention exactly where in the revised manuscript this change can be found – page 8,

1 paragraph, and 214-220(Figure3).]

Figure 3. Nomogram model for predicting AVF dysfunction risk.

Sig: The nomogram was used to predict the risk of AVF dysfunction in MHD patients. For each variable (PLT abnormal, platelet abnormal; TC, total cholesterol; (Ca-P) product count abnormal, calcium–phosphorus product abnormal; TP, total protein; Diabetes; Fibrinogen abnormal), a vertical line was drawn upward from its value to determine an individual score. The scores of all variables were summed to obtain a total score. A vertical line was then drawn downward from this total score to the risk axis, identifying the patient's predicted risk of AVF dysfunction.

3.Mention exactly where in the revised manuscript this change can be found – page 9,1 paragraph, and 237-243(Figure 4).]

Figure 4: AUC of the training and validation sets.

Sig:The Area Under the Curve (AUC) quantifies model discrimination by summarizing the overall performance of the ROC curve. A perfect classifier achieves an AUC of 1, corresponding to the top-left corner of the plot, while an AUC of 0.5 indicates predictions no better than random chance. Furthermore, the ROC curve itself plots the True Positive Rate against the False Positive Rate across classification thresholds and enables identification of optimal operating points through explicit visualization of the sensitivity-specificity tradeoff.

4.Mention exactly where in the revised manuscript this change can be found – page10,

1 paragraph, and 252-250(Figure 5).]

Figure 5: Calibration curve of AVF dysfunction prediction model.

Sig: Calibration plots ofthe predictive model. A Callbration curve of the model in the training set; B Calibration curve of the model in the internal validation set; The calibration curve displays the predicted probability of AVF dysfunction on the x-axis and the actual diagnosed risk of AVF dysfunction on the y-axis. The diagonal dashed line in the graph represents the ideal prediction model, while the solid black line represents the risk nomogram prediction model developed in this study. When the solid black line closely approximates the diagonal dashed line, it indicates better predictive performance of the model. As shown in the graph, the developed risk nomogram prediction model for AVF dysfunction demonstrates good predictive accuracy.

5.Mention exactly where in the revised manuscript this change can be found – page11,

1 paragraph, and 272-275(Figure 6).]

Figure 6. Decision curve analysis for AVF dysfunction prediction model.

Sig: (C) Decision curve of the test set. (D) Decision curve of the validation set. The decision tree model demonstrates superior net benefit across a threshold probability range of 0.15 to 0.78, as its DCA curve (red) remains above the "ALL" (grey) and "NONE" (black) reference strategies. The "ALL" model classifies all cases as positive, while the "NONE" model classifies all as negative.

Comments 2: Auhtors refering to the supplementary, figure legend detail should be in the main manuscript file with the figure. 

Response 2:Thank you for pointing this out. We agree with this comment and improve and revise the entire article with the modifications highlighted in yellow .

1. Mention exactly where in the revised manuscript this change can be found – page 6-7, and 177-204.]

 3.2 Collinearity assessment

Based on the univariate analysis results (P<0.02), variables meeting the significance threshold were selected for collinearity assessment. Using linear regression models, multicollinearity was evaluated through variance inflation factor (VIF) and Tolerance values, with VIF < 10 and Tolerance > 0.1 defined as acceptable thresholds. Finally, seven key predictors (TC, total protein, diabetes, hs-crp, PLT, fibrinogen, serum (ca-p) product, and hypotension) were identified by integrating statistical collinearity diagnostics with clinical relevance in Table 2.

Table 2. Multicollinearity diagnostic of logistic regression

Item

Tolerance

VIF

Hypotension after dialysis

0.947

1.056

Fibrinogen abnormal

0.968

1.033

(Ca-P) product count abnormal

0.980

1.021

PLT abnormal

0.989

1.012

Complicated diabetes

0.933

1.072

TP(g/L)

0.995

1.006

TC(mmol/L)

0.972

1.029

Sig: PLT, platelet; TC, total cholesterol; Ca-P, serum calcium–phosphorus product; TP, total protein.

3.3. Analysis of risk factors for arteriovenous fistula dysfunction in the modeling group

Seven key predictors (abnormal PLT, TC, diabetes , abnormal Fib, abnormal (Ca-P) ,  hypotension after dialysis,TP) were identified as independent factors for AVF dysfunction among MHD patients by integrating statistical results from univariate and collinearity analyses with clinical relevance, a binary logistic regression analysis was conducted. The assignment of the independent variable values is presented in Table 3. The results indicated that abnormal PLT (OR=4.18,95%CI:1.95-8.99), TC (OR=1.71,95%CI:1.31-2.24), diabetes (OR=3.20,95%CI:1.77-5.77), abnormal Fib (OR=2.65,95%CI:1.46-4.80), abnormal (Ca-P) (OR=1.81,95%CI:1.00-3.27), and hypotension after dialysis (OR=4.14,95%CI:2.31-7.42). TP (OR=0.93,95%CI:0.89-0.97) was a protective factor for AVF dysfunction among MHD patients according to multivariate logistic regression analysis(P<0.05 Table 3-4 and Figure 2).

Table 3. The assignment of logistic regression

Item

Assignmen

Hypotension after dialysis

1: Yes; 0: No

Fibrinogen abnormal

1: Yes; 0: No

(Ca-P) product count abnormal

1: Yes; 0: No

PLT abnormal

Continuity variables

Complicated diabetes

1: Yes; 0: No

TP(g/L)

Continuity variables

TC(mmol/L)

Continuity variables

Sig: PLT, platelet; TC, total cholesterol; Ca-P, serum calcium–phosphorus product; TP, total protein.

Table 4. Multi-factor logistic regerssion analysis of risk factors of AVF dysfunction

β

SE

Wald

P

OR

95% CI

Hypotension after dialysis

1.421

0.298

22.765

0.001

4.141

2.310-7.422

Fibrinogen (g/L)

0.937

0.304

10.217

0.001

2.645

1.457-4.802

(Ca-P) product count

0.594

0.301

3.899

0.048

1.812

1.004-3.267

PLT(x109/L)

1.431

0.390

13.433

0.001

4.182

1.946-8.988

Complicated diabetes

1.161

0.302

14.810

0.001

3.194

1.768-5.772

TP(g/L)

-0.076

0.022

11.597

0.001

0.927

0.887-0.968

TC(mmol/L)

0.538

0.136

15.570

0.001

1.712

1.311-2.237

Sig: PLT, platelet; TC, total cholesterol; Ca-P, serum calcium–phosphorus product; TP, total protein; SE, standard error; OR, odds ratio; CI, confidence interval.

Figure 3. Nomogram model for predicting AVF dysfunction risk.

Sig: The nomogram was used to predict the risk of AVF dysfunction in MHD patients. For each variable (PLT abnormal, platelet abnormal; TC, total cholesterol; (Ca-P) product count abnormal, calcium–phosphorus product abnormal; TP, total protein; Diabetes; Fibrinogen abnormal), a vertical line was drawn upward from its value to determine an individual score. The scores of all variables were summed to obtain a total score. A vertical line was then drawn downward from this total score to the risk axis, identifying the patient's predicted risk of AVF dysfunction.

3.Mention exactly where in the revised manuscript this change can be found – page 9,1 paragraph, and 237-243(Figure 4).]

Figure 4: AUC of the training and validation sets.

Sig:The Area Under the Curve (AUC) quantifies model discrimination by summarizing the overall performance of the ROC curve. A perfect classifier achieves an AUC of 1, corresponding to the top-left corner of the plot, while an AUC of 0.5 indicates predictions no better than random chance. Furthermore, the ROC curve itself plots the True Positive Rate against the False Positive Rate across classification thresholds and enables identification of optimal operating points through explicit visualization of the sensitivity-specificity tradeoff.

4.Mention exactly where in the revised manuscript this change can be found – page10,1 paragraph, and 252-250(Figure 5).]

Figure 5: Calibration curve of AVF dysfunction prediction model.

Sig: Calibration plots ofthe predictive model. A Callbration curve of the model in the training set; B Calibration curve of the model in the internal validation set; The calibration curve displays the predicted probability of AVF dysfunction on the x-axis and the actual diagnosed risk of AVF dysfunction on the y-axis. The diagonal dashed line in the graph represents the ideal prediction model, while the solid black line represents the risk nomogram prediction model developed in this study. When the solid black line closely approximates the diagonal dashed line, it indicates better predictive performance of the model. As shown in the graph, the developed risk nomogram prediction model for AVF dysfunction demonstrates good predictive accuracy.

5.Mention exactly where in the revised manuscript this change can be found – page11,1 paragraph, and 272-275(Figure 6).]

Figure 6. Decision curve analysis for AVF dysfunction prediction model.

Sig: (C) Decision curve of the test set. (D) Decision curve of the validation set. The decision tree model demonstrates superior net benefit across a threshold probability range of 0.15 to 0.78, as its DCA curve (red) remains above the "ALL" (grey) and "NONE" (black) reference strategies. The "ALL" model classifies all cases as positive, while the "NONE" model classifies all as negative.

Comments 2: Auhtors refering to the supplementary, figure legend detail should be in the main manuscript file with the figure. 

Response 2:Thank you for pointing this out. We agree with this comment and improve and revise the entire article with the modifications highlighted in yellow .

1. Mention exactly where in the revised manuscript this change can be found – page 6-7, and 177-204.]

 3.2 Collinearity assessment

Based on the univariate analysis results (P<0.02), variables meeting the significance threshold were selected for collinearity assessment. Using linear regression models, multicollinearity was evaluated through variance inflation factor (VIF) and Tolerance values, with VIF < 10 and Tolerance > 0.1 defined as acceptable thresholds. Finally, seven key predictors (TC, total protein, diabetes, hs-crp, PLT, fibrinogen, serum (ca-p) product, and hypotension) were identified by integrating statistical collinearity diagnostics with clinical relevance in Table 2.

Table 2. Multicollinearity diagnostic of logistic regression

Item

Tolerance

VIF

Hypotension after dialysis

0.947

1.056

Fibrinogen abnormal

0.968

1.033

(Ca-P) product count abnormal

0.980

1.021

PLT abnormal

0.989

1.012

Complicated diabetes

0.933

1.072

TP(g/L)

0.995

1.006

TC(mmol/L)

0.972

1.029

Sig: PLT, platelet; TC, total cholesterol; Ca-P, serum calcium–phosphorus product; TP, total protein.

3.3. Analysis of risk factors for arteriovenous fistula dysfunction in the modeling group

Seven key predictors (abnormal PLT, TC, diabetes , abnormal Fib, abnormal (Ca-P) ,  hypotension after dialysis,TP) were identified as independent factors for AVF dysfunction among MHD patients by integrating statistical results from univariate and collinearity analyses with clinical relevance, a binary logistic regression analysis was conducted. The assignment of the independent variable values is presented in Table 3. The results indicated that abnormal PLT (OR=4.18,95%CI:1.95-8.99), TC (OR=1.71,95%CI:1.31-2.24), diabetes (OR=3.20,95%CI:1.77-5.77), abnormal Fib (OR=2.65,95%CI:1.46-4.80), abnormal (Ca-P) (OR=1.81,95%CI:1.00-3.27), and hypotension after dialysis (OR=4.14,95%CI:2.31-7.42). TP (OR=0.93,95%CI:0.89-0.97) was a protective factor for AVF dysfunction among MHD patients according to multivariate logistic regression analysis(P<0.05 Table 3-4 and Figure 2).

Table 3. The assignment of logistic regression

Item

Assignmen

Hypotension after dialysis

1: Yes; 0: No

Fibrinogen abnormal

1: Yes; 0: No

(Ca-P) product count abnormal

1: Yes; 0: No

PLT abnormal

Continuity variables

Complicated diabetes

1: Yes; 0: No

TP(g/L)

Continuity variables

TC(mmol/L)

Continuity variables

Sig: PLT, platelet; TC, total cholesterol; Ca-P, serum calcium–phosphorus product; TP, total protein.

Table 4. Multi-factor logistic regerssion analysis of risk factors of AVF dysfunction

β

SE

Wald

P

OR

95% CI

Hypotension after dialysis

1.421

0.298

22.765

0.001

4.141

2.310-7.422

Fibrinogen (g/L)

0.937

0.304

10.217

0.001

2.645

1.457-4.802

(Ca-P) product count

0.594

0.301

3.899

0.048

1.812

1.004-3.267

PLT(x109/L)

1.431

0.390

13.433

0.001

4.182

1.946-8.988

Complicated diabetes

1.161

0.302

14.810

0.001

3.194

1.768-5.772

TP(g/L)

-0.076

0.022

11.597

0.001

0.927

0.887-0.968

TC(mmol/L)

0.538

0.136

15.570

0.001

1.712

1.311-2.237

Sig: PLT, platelet; TC, total cholesterol; Ca-P, serum calcium–phosphorus product; TP, total protein; SE, standard error; OR, odds ratio; CI, confidence interval.

Reviewer 4 Report

Comments and Suggestions for Authors

The authors have revised the manuscript thoroughly by addressing all the comments provided by the reviewers. The paper has significantly improved in quality and is now suitable for publication in its present form.

Author Response

1.Comments 1: The authors have revised the manuscript thoroughly by addressing all the comments provided by the reviewers. The paper has significantly improved in quality and is now suitable for publication in its present form.

Response 1:I am truly grateful for your recognition and affirmation.
